# Molecularly Imprinted Polymer-Based Biomimetic Systems for Sensing Environmental Contaminants, Biomarkers, and Bioimaging Applications

**DOI:** 10.3390/biomimetics8020245

**Published:** 2023-06-08

**Authors:** Kalaipriya Ramajayam, Selvaganapathy Ganesan, Purnimajayasree Ramesh, Maya Beena, Thangavelu Kokulnathan, Arunkumar Palaniappan

**Affiliations:** 1Department of Chemistry, School of Advanced Sciences, Vellore Institute of Technology (VIT), Vellore 632014, Tamil Nadu, India; kalaipriya.r@vit.ac.in (K.R.); selvaganapathy.g@vit.ac.in (S.G.); 2Centre for Biomaterials, Cellular and Molecular Theranostics (CBCMT), Vellore Institute of Technology (VIT), Vellore 632014, Tamil Nadu, India; purnimajayasrees.r@vit.ac.in (P.R.); maya.b2020@vitstudent.ac.in (M.B.); 3School of Biosciences and Technology, Vellore Institute of Technology (VIT), Vellore 632014, Tamil Nadu, India; 4Department of Electro-Optical Engineering, National Taipei University of Technology, Taipei 106, Taiwan

**Keywords:** MIP, imprinting, sensors, electrochemical, optical, bioimaging

## Abstract

Molecularly imprinted polymers (MIPs), a biomimetic artificial receptor system inspired by the human body’s antibody-antigen reactions, have gained significant attraction in the area of sensor development applications, especially in the areas of medical, pharmaceutical, food quality control, and the environment. MIPs are found to enhance the sensitivity and specificity of typical optical and electrochemical sensors severalfold with their precise binding to the analytes of choice. In this review, different polymerization chemistries, strategies used in the synthesis of MIPs, and various factors influencing the imprinting parameters to achieve high-performing MIPs are explained in depth. This review also highlights the recent developments in the field, such as MIP-based nanocomposites through nanoscale imprinting, MIP-based thin layers through surface imprinting, and other latest advancements in the sensor field. Furthermore, the role of MIPs in enhancing the sensitivity and specificity of sensors, especially optical and electrochemical sensors, is elaborated. In the later part of the review, applications of MIP-based optical and electrochemical sensors for the detection of biomarkers, enzymes, bacteria, viruses, and various emerging micropollutants like pharmaceutical drugs, pesticides, and heavy metal ions are discussed in detail. Finally, MIP’s role in bioimaging applications is elucidated with a critical assessment of the future research directions for MIP-based biomimetic systems.

## 1. Introduction

Precise molecular recognition of the analytes paired with advanced techniques to monitor those changes in the recognition elements is currently being explored to fabricate highly sensitive and specific biosensors. Precise molecular recognition, such as receptor-ligand interactions, antibody-antigen complex formation, and enzyme-substrate reactions, is ubiquitous in biology and performs many complex functions within cells or during cell-cell communications. Such meticulous molecular recognition systems are widely explored in the fabrication of biosensors. However, these natural recognition components exhibit inherent limitations, including high cost, limited stability, and batch-to-batch variations. For example, while considering all antibodies in the market, it has been stated that 75% of antibodies have not been validated or do not perform adequately for the application [1]. Using animals (and subsequent animal sacrifice) in conventional antibody manufacturing raises further ethical issues. There is still a significant reliance on animal-derived antibodies despite advancements in validation strategies and significant industry expenditure. There is a significant push to develop alternatives to antibodies because it is estimated that one million animals are used annually in Europe alone to produce antibodies. The European Union (EU) Reference Laboratory issued a new recommendation on non-animal-derived antibodies in 2020, which calls for substituting animal-derived antibodies where possible and is anticipated to have a significant impact on the future of antibody production in the EU [2]. Using more stable, smaller counterparts for natural receptors is one way to replace them. Despite having a different structural form from antibodies, these smaller counterparts are known as “antibody mimics” because they perform similar tasks. Unfortunately, these antibody mimics are costly and have limited market availability, probably because there is no platform technology for purification [3]. An example of antibody mimics includes single-chain variable fragments (scFvs) and fusion proteins from the variable sections of the heavy and light chains of immunoglobulins connected via a short linker peptide [4]. Fab fragments (antigen-binding fragments) are composed of the whole light chain and the variable region of the heavy chain of an antibody and have the advantage of being inexpensive and straightforward to develop (it takes a few days) for sensing applications. In contrast, scFvs have the advantage of being highly customizable, which will increase sensitivity. Although these fragments can denature when immobilized on sensor surfaces, synthetic recognition elements often exhibit superior specificity [5]. Aptamers are single-stranded peptide or oligonucleotide molecules that fold into definite structural designs and, therefore, can bind specifically and selectively to target molecules. However, the aptamer’s binding affinity is poor compared to the monoclonal antibodies [6]. Thus, aptamers are not much preferred for translational applications.

There is a need to develop synthetic molecular recognition units that mimic natural molecular recognition systems and biomimetic molecular recognition systems. One such biomimetic system is the MIPs, polymeric recognition elements that follow a similar pattern of mimicking antibodies [7]. MIPs are a group of customizable analogs that replicate the natural interactions between an antibody, an antigen, an enzyme, and a substrate. The specific recognition site in the MIPs depends on the “molecular lock and key” mechanism, which Emil Fischer postulated selectively binds the active site present in the template molecules [8]. Because of their good qualities, such as robustness, stability, ease of manufacture, high affinity, and selectivity towards the target molecule, MIPs have drawn the attention of scientists [9,10,11].

Successful interaction between the recognition site and the requisite template is made possible by various binding modalities such as covalent [12], semi-covalent [13], and non-covalent bonding [14]. The selectivity of MIPs is equivalent to, and in some cases even superior to, that of conventional analytical techniques. These substances are a cost-effective substitute that frequently enables the quantitative on-site assessment of analytes.

MIPs have been extensively used for solid phase extraction [15,16], chromatographic separation [17], catalysis [18], drug delivery [19], protein binding [20], environmental and biomedical sensing [21], water and wastewater treatment [22], and membrane-based separations [23]. The most widely used application of MIPs, notably in analytical chemistry, is purification [24]. The potential of MIP-based sensors for environmental and biomedical applications to detect compounds at trace levels in complex matrices without pre-treatment opens possibilities for in-situ contamination monitoring and quick clinical analysis at the point of care for better diagnosis and treatment. Most MIP-based technology has remained in the academic world, despite a real market need for such devices. In this review, various polymerization and imprinting techniques for MIPs are elaborated on in detail. Furthermore, recent advancements in MIPs-based optical and electrochemical sensors for detecting environmental pollutants and biomarkers are reviewed in detail. In the later part, MIPs-based bioimaging applications are also reviewed with the author’s perspectives on the future directions of MIPs in these research areas, and challenges in the field are explained.

## 2. Preparation Methods for MIPs

Traditionally, MIPs are prepared through the polymerization of monomers using free radicals generated through several fabrication methods such as bulk, precipitation, emulsion, suspension techniques, and others. Figure 1 gives a schematic illustration of various polymerization techniques used to synthesize IPs [25]. Table 1 lists the most used functional monomers, initiators, crosslinkers, and porogens for the synthesis of MIPs.

### 2.1. Bulk Polymerization

Bulk polymerization, a conventional method for synthesizing a monolith or block MIPs, requires a template, functional monomers, cross-linker, initiator, and porogen in a non-polar solvent in a specific proportion. Photo- or thermal energy is used to initiate the polymerization process. The resulting polymers were ground and sieved to break them, and subsequently, the template was extracted using eluents [36,37].

### 2.2. Suspension Polymerization

Suspension polymerization is a simple, one-step radical polymerization process that results in the formation of spherical MIPs. Here, the functional monomers, initiators, template molecules, and porogen are mixed to form a dispersed phase that is then suspended onto an aqueous phase (continuous phase) as droplets [37]. In the constant phase, polyvinyl alcohol is mainly used as the suspending agent (substances added in a colloidal system to prevent aggregation of particles and thus keep them suspended longer in the continuous phase) for enhancing stability. The amount of porogen employed in the process can efficiently control the particles’ porosity on the surface [38,39].

### 2.3. Emulsion Polymerization

Emulsion polymerization is also a radical polymerization technique where polymerization occurs inside the micelles formed in the oil/water (O/W) emulsion system. The O/W emulsion system is developed using monomer droplets (as oil) emulsified onto a continuous water phase of surfactant molecules. Although suspension and emulsion polymerization appear to be similar, the critical difference in the process is that the initiator used in emulsion polymerization is water-soluble and thus must enter the micelle for the polymerization reaction. However, in suspension polymerization, the initiator molecules are soluble in oil (monomer) and react with the monomer molecules, resulting in spherical MIPs. Emulsion polymerization results in spherical nanoparticles of size 10–100 nm [38,39,40].

### 2.4. Precipitation Polymerization

Precipitation polymerization involves the polymerization of monomers using an initiator, both dissolved in a solvent without stabilizers or additives, resulting in the precipitation of spherical-shaped MIPs [41,42].

### 2.5. Multi-Step Swelling Polymerization

Multi-step swelling polymerization, also known as seed polymerization, involves polymerizing preformed monodispersed seeds (which contain a pre-polymerization mixture) to obtain uniform spherical particles [43]. A suitable organic solvent is added to initiate swelling of the seeds to reach a desired size of 5–10 μm, after which polymerization is induced by the addition of required constituents such as monomers and initiators. MIPs fabricated through multiple-step swelling are ideal for chromatographic applications. This complex and complicated method requires specific reaction conditions [44,45,46].

### 2.6. Surface Imprinting Polymerization

The conventional polymerization techniques discussed above resulted in bulk polymers of various shapes and sizes based on the process parameters. However, the complete removal of template molecules from the whole MIPs, especially from the interior of particles, required vast amounts of solvents, which negatively influenced the adsorption capacity and stability of the MIPs [38]. To overcome these drawbacks, surface imprinting polymerization has been developed that involves grafting a thin layer of MIP onto the surface of carriers, sometimes beads such as porous silica or spherical polymers. Several types of spherical MIPs can also be prepared with these possibilities [39]. After polymerization, the core silica particles are etched away, leaving only MIPs.

### 2.7. Electrochemical Polymerization

Electrochemical polymerization, or electro-polymerization, is a technique based on the deposition of MIPs onto the surface of electrode material in the presence of a template. The polymerization setup consists of three electrodes: (a) the working electrode: the electrode in which deposition of MIPs takes place; (b) the reference electrode: typically Ag/AgCl or saturated calomel electrode, SCE; and (c) a counter electrode: platinum or nickel electrodes. These electrodes are immersed in an electrochemical cell containing electrolyte solution, electroactive monomers, templates, and solvents. Upon application of a potential to the working electrode, the monomers are electrochemically oxidized to produce free radicals, which initiate the polymerization process to form a conductive or non-conducting polymeric film on the surface of the working electrode. Notably, the 3-aminophenyl boronic acid (APBA), pyrrole (Py), polythiophene (PTh), and aniline (ANI)-based electroactive monomers are polymerized to produce conductive MIPs [41,42,47,48,49]. Monomers like phenol (Ph), 1,2-phenylenediamine (PD), and thiophenol (TPh) are electropolymerized for non-conducting MIPs [44,45,50,51]. Nonconductive MIP films are preferably used for capacity chemosensors, and conductive polymer films are applied for electrochemical sensor studies [52,53]. This process can be easily achieved by various electrochemical techniques, namely voltametric [54], potentiostatic [55], and galvanostatic [56].

Voltammetric polymerization is the most popular fabrication route for electropolymerization. Cyclic voltammetry is one such technique in which potentials are varied, which leads to the oxidation of monomers and the deposition of MIPs on the working electrode. The voltage range can be varied to optimize the thickness of the MIP film. The potentiostatic route of electro-polymerization takes place by applying a constant potential. Thus, identification of this potential (here, the potential is fixed based on the results from the voltammetric analysis) is crucial for controlling the thickness, stability, and conductivity of the MIP film formed upon the electrode surface. The galvanostatic electro-polymerization technique is similar to that of the potentiostatic method. However, this depends on the application of a constant current to induce the polymerizationprocess [57]. The advantages and limitations of the MIP polymerization methods are given below in Table 2.

## 3. Imprinting Techniques for MIPs

A molecular imprinting approach depends on the interactions between the template molecule and the functional monomer, which can be either covalent, non-covalent, or semi-covalent, as depicted in Figure 2 [62,63].

### 3.1. Covalent Imprinting

Covalent imprinting forms covalent bonds between the template and the monomer for fabricating MIPs. This method, also known as stoichiometric imprinting, involves MIPs exhibiting homogeneous cavity distribution and minimal incomplete binding sites, thus possessing enhanced selectivity. In 1977, Wulff and his research group developed the first MIPs utilizing this strategy by copolymerizing 4-nitrophenyl *x*-mannopyranoside-2,3;4,6-di-*O*-(4-vinyl phenyl boronate) with ethylene methacrylate and methyl methacrylate [12]. However, only limited reactions form covalent bonds between the template and the functional monomer, which are reversible under mild conditions, resulting in slow analyte binding and unbinding. Using this method, a reversible covalent binding condensation reaction is used to link the polymerizable molecule with the imprinted molecule via ketal [64], acetal [61], esters [65], boronate [66], and Schiff base bonds [67]. The functional monomer and the template must be broken apart by acid hydrolysis [68]. However, this imprinting technique is limited to functional monomers and templates such as alcohols, amines, ketones, aldehydes, or carboxylic acids [69].

### 3.2. Non-Covalent Imprinting

Non-covalent imprinting involves the development of non-covalent interactions, such as hydrogen bonding, van der Waals forces, π-π and hydrophobic interactions, electrostatic forces, and metal coordination, both in MIP synthesis and analyte binding and unbinding. In 1981, Mosbach and colleagues introduced non-covalent imprinting of L-phenylalanine-anilide using MAA as the functional monomer. They used ionic interactions, hydrophobic interactions, hydrogen bonding, and charge transfer interactions between the template and monomers [70]. One of the most important and extensively used monomer-crosslinker systems for non-covalent imprinting techniques includes combining MAA as a functional monomer and ethylene glycol dimethacrylate (EGDMA) as a crosslinking agent, in which MAA can form hydrogen bonds between varieties of template molecules. The broad usage of MAA as a functional monomer is due to its ability to interact with various functional groups, like esters, acids, amides, and amine substituents. This method uses more monomer templates to generate enough interaction sites. In this strategy, the electrostatic force dominates, and other forces, such as hydrogen bonding, support improving the recognition properties [71].

### 3.3. Semi-Covalent Interactions

Semi-covalent imprinting combines covalent and non-covalent imprinting, in which the template-monomer complex is formed by covalent interactions and analyte binding by noncovalent interactions [60]. In 1990, the first semi-covalent approach was reported by Sellergren and Anderson. These strategies employ covalent template-monomer complexes in the imprinting step but entirely non-covalent interactions (electrostatic and hydrogen bonding interactions) for analyte binding [72]. This type of interaction is employed in various other systems reported in the literature [73,74,75].

### 3.4. Metal-Mediated Interactions

The number of applications for MIPs is increasing due to the combination of molecular imprinting with metal ions. During pre-polymerization, metal ions facilitate interactions between the monomer and the template molecule, forming ionic bonds rather than weaker hydrogen ones [76]. Various strategies have been used for the fabrication of metal-ion imprinted polymers: (i) crosslinking with a bifunctional reagent of linear chain polymers having metal-binding ligands; (ii) copolymerization of metal complexes containing polymerizable ligands with a cross-linker; and (iii) surface imprinting at the interface of water-in-oil emulsions through assembly with amphiphilic functional monomers. In this case, the translational metal ion is complexed by polymerizable ligands and the target molecule, which can be a neutral or charged species. The metal ion’s charge and ligand characteristics can influence the strength of the interactions. The polymer obtained via this approach can be employed for various applications, which include ion-selective sensors [72,73], catalytic applications, etc. [75].

## 4. MIPs-Based Sensors

### 4.1. MIP-Based Optical Sensors

MIP-based optical sensors are known for their simplicity, ease of manufacture, and ability to achieve a very low detection limit. MIP-based optical sensors contain MIPs as the recognition unit to interact and bind significantly with the desired target analyte and as the transducer component for signaling the binding event. MIPs in the sensors typically enhance their specificity by binding specifically to the targeted analyte of interest. Furthermore, various types of MIP-based optical sensors, such as fluorescence, colorimetric, surface plasmon resonance, and surface-enhanced Raman scattering (SERS), have made considerable advancements in recent years to detect toxic pollutants as well as in bio-sensing applications [77]. These sensors use the principles of change in light intensities [73,76]: (i) signal to turn off, or (ii) signal to turn on [78], which is depicted in Figure 3 and Figure 4. For instance, in fluorescence-based sensors, the specific binding of fluorescentMIPs with target analytes resulted in either enhancement or quenching of the fluorescent signal, resulting in the detection and further quantification of analytes [79].

MIP-based colorimetric sensors have emerged as a potential cost-effective analytical tool for analyte detection based on the color changes due to the specific interaction with the analytes of interest [80]. In recent years, significant efforts have been made to improve the properties of optical sensors by modifying or adding components like quantum dots (QDs). Wang et al. and Yang et al. developed a fluorescent sensor based on quantum dots material integration with MIPs, which improves the properties of QDs-MIPs, such as binding kinetics, selectivity, sensitivity, and reliability. No cross-reactivity was observed against other structural analogs, and it was also verified that there was no competition for the binding sites in the presence of potentially interfering or competing species. LOD values in the range of μg L^−1^ were achieved using this method [81,82]. MIPs-based optical sensors in different environmental applications are listed in Table 3.

#### 4.1.1. MIPs-Based Optical Sensors for Pharmaceutical Drug Detection

MIPs are currently being used to detect a wide gamut of analytes (proteins, drugs, biomolecules, etc.) and several proteomic analyses using surface plasma resonance [111]. He et al. demonstrated an optical fiber-based sensor fabrication method for detecting dabrafenib (an anticancer drug) using ELISA. They used methacrylate alkoxy silane as a monomer to synthesize MIPs and found that the detection limit was 74.4 μgmL^−1^. Furthermore, the sensor showed selectivity toward the drug, thereby confirming the selectivity of MIP toward the drug [44]. Similarly, in another study, Altintas et al. synthesized MIP (silica)-based nanoparticles with high affinity for diclofenac and were found to detect 1.24–80 ng mL^−1^, confirming the potency for the detection of diclofenac in water through a UV-Vis spectrophotometer [112]. Aurelio et al. also used a sensor based on an optical fiber long-period grating MIP to detect cocaine. They used nanoparticle-based MIPs with a detection limit of 0.24 ng mL^−1^ without cross-reacting with morphine, confirming the high specificity and sensitivity, which were confirmed using ELISA and qPCR [44]. Wang et al. developed a dual emission (carbon and CdTe) QDs-based MIP method to detect dopamine in biofluid. Dual emission is due to combining two different quantum dots with different color emissions (i.e., red, and blue) through a molecular imprinting process. Specifically, the blue-emission QDs were embedded in silica nanocores to maintain constant fluorescence intensity. In contrast, the red-emission QDs were mixed into the imprinted polymer shell, thus enabling interaction with dopamine molecules to induce fluorescence quenching during dopamine recognition. This way, dopamine is observed using a paper-based colorimetric method (Figure 5) [113]. A representation of a nanoparticle-based optical sensor with high detection potential. Even though MIP-based detection demands cost-effectiveness and increased stability, it also should have selective recognition and biocompatibility. More studies on this line need to be explored to prove MIP as a promising biomedical device and make these prospects a reality [114].

#### 4.1.2. MIPs in the Detection of Bacteria and Viruses

MIPs have been found to recognize viruses and bacteria, which could be further utilized to control and prevent viral and bacterial infections. Infectious diseases caused by *E. coli*, *P. aeruginosa*, *L. monocytogenes*, *S. paratyphi*, *P. mirabilis,* and many more are significant concerns for public health [115]. Several traditionally used techniques effectively detect bacteria and viruses, including PCR-based approaches, ELISA, and many others. However, ELISA-based procedures are costly, time-consuming, and labor-intensive and require skilled personnel and expensive equipment, allowing room for MIP-based biosensor tools in the healthcare field [116]. Furthermore, these biosensors are sensitive and less time-consuming than other traditional methods.

Moreover, biosensors based on molecular imprinting technology effectively detect bacteria and viruses [117]. Tokonami et al. fabricated an oxidized MIP polypyrrole film as a highly selective and rapid detection system for *P. aeruginosa* even in a mixture of bacterial cultures containing *Acinetobacter calcoaceticus*, *E. coli*, and *Serratia marcescens* with a LOD of 10^3^ to 10^9^ CFU/mL, which was analyzed using di-electrophoresis [118]. Hong et al. developed an immune-like membrane to isolate and detect C-reactive protein in serum samples using MIP-based nanocavities. They improved the performance of isolation by aligning the C-reactive protein. C-reactive protein (CRP) is a sensitive marker of inflammation. It is primarily synthesized in hepatocytes in response to proinflammatory cytokines, such as tumor necrosis factor-alpha and interleukin 6, because of acute or chronic stimuli [119]. They fabricated and demonstrated the adhesion forces of the MIP-based nanocavities on immune-like membranes and integrated them with microfluidic systems as point-of-care applications (Figure 6) [120].

Viral and bacterial infections spread extensively, at a faster rate. Appropriate diagnosis and treatment strategies are necessary for better prevention and cure [111]. Here, Cennamo et al., Bognar et al., and Ayankojo et al. demonstrated an acrylamide-based MIP-coated gold chip, which has specific recognition towards the subunits of the SARS-CoV-2 protein, with a higher sensitivity and faster response, which was confirmed using a spectrophotometer [117,118,119]. In another study, Zhangab et al. prepared a magnetic resonance light scattering sensor based on virus magnetic MIP nanoparticles (effective concentration of 90 ng mL^−1^) to detect hepatitis. A virus and the subsequent capture of this virus onto the particle’s surface upon application of the magnetic field. The sensor was able to detect a deficient concentration of virus in the picomolar (pM) range (low detection limit of 6.2 pmol L^−1^) [21]. However, thorough investigations are still needed to improve the selectivity and potential of shape recognition in sensors based on MIPs. Tawfik et al. developed fluorescent molecularly imprinted conjugated polythiophene nanofibers (FMICP NFs) paper-based devices, which have an enzyme-free signal-amplification capability for AFP (alpha-fetoprotein) biomarker detection (Figure 7) [121].

#### 4.1.3. MIPs in Bioimaging

Bioimaging is vital in bioscience since it allows for targeting, localizing, and visualizing biological activities in cells or tissues [122]. MIPs, in combination with QDs, have been widely explored for bioimaging applications for the past few years. This combination has gained attention over antibodies due to their high stability, low cost, long shelf life, etc. [123]. Furthermore, MIPs are known for their low immunogenic response, specificity to the target area, and ability to cross the cell membrane. QDs used with MIPs are highly biocompatible and, due to this, have been used as a powerful tool for bioimaging purposes. Cecchini et al. showed nano-MIPs synthesized from nine amino acid surface epitopes of h-VEGF to detect human vascular endothelial growth factor in human melanoma tumors by binding to the protein (VEGF) specifically and helping in localizing progressive tumor cells with green fluorescencein vivo [124]. Peng et al. also demonstrated a method of developing a theranostic device with improved therapeutic efficiency that contains gadolinium-doped silicon quantum dots with MIPs for cancer (MCF-7) detection through MRI and fluorescence imaging. These molecules produced reactive oxygen species upon laser irradiation using a 655 nm laser (300 mW/cm^2^) for 10 min, killing the cancer cells in the mouse tumor models [125]. Wang et al. and Yet et al. also designed FITC-doped SiO_2_ nanoparticles that imprinted MIPs with HER2-glycan (MIP) for imaging hepatic carcinoma and breast cancer cells. This study confirmed that monosaccharide particles enabled efficient stability and specificity to the target area at a concentration of 200 µg/mL [121,122]. Thesein vitro andin vivoexperiments confirmed the exceptional tumor-targeting capability and specificity of the MIP-conjugated moieties, thereby making them a translatable approach for cancer therapy (Figure 8) [126].

## 5. MIP-Based Electrochemical Sensor

Electrochemical sensors are among the most common to detect environmental pollutants and biological analytes due to their high sensitivity, better LOD, economics, and portability [108,127]. Electrochemical sensors consist of a cell with a working electrode of particular interest accompanied by a reference electrode and an auxiliary electrode [16,109]. They are classified into three categories: capacitance sensors, voltammetric sensors, and potentiometric sensors based on their measured electrochemical parameters. The capacitance sensor measures the change in conductivity over time as a function of the target concentration. The voltammetric sensor measures the target’s effect on the redox reaction’s current potential. The potentiometric sensor measures the potential of the redox reaction to measure the concentration [16,109].

### 5.1. MIP-Based Electrochemical Sensor in Environmental Applications

The detection and quantification of pollutants present in the environment are necessary to determine their fate and transport. Recently, Mehmandoust et al. developed a MIP film-loaded metal-organic framework (MOF) to detect fenamiphos (an insecticide) in vegetables using gold-doped graphitic carbon nitride nanosheets. A LOD value of 7.13 nM was achieved with a satisfactory recovery of 94.7–107.9% [128]. MIP-based electrochemical sensors for detecting different environmental pollutants are listed in Table 4.

Ghaani et al. developed a MIP-based novel electrochemical sensor for the detection of 4,4′-methylene diphenyl diamine (MDA), a primary aromatic amine typically used in the preparation of polyurethane foams but found to have carcinogenic properties—electrodeposition-coated MIP with multiwalled carbon nanotube (MWCNT) on glassy carbon electrodes. The MWCNT improved the sensor’s sensitivity through its antifouling properties. Furthermore, the different parameters, such as incubation time, scan cycles, elution time, pH, and molar ratio of template molecules to monomers, were optimized to enhance the sensor’s sensitivity with a final LOD value of 15 nM. The actual sample analysis of MDA was performed, and the recovery rate was between 94.10% and 106.76% [129]. Zhou et al. designed the gold nanoparticles/reduced graphene oxide (AuNP/RGO) modified MIP sensor for the selective detection of nitrofurazone (an antibiotic drug). In this, o-phenylenediamine (*o*–PD) was used as a functional monomer in MIP preparation. Here, the differential pulse voltammetry (DPV) technique was used to detect nitrofurazone by redox probe ([Fe (CN)_6_]^3−/4−^) with a low detection limit of 0.18 nmol L^−1^ and also produced a reasonable recovery rate of 99.06–101.46% inaccurate water analysis [130]. A cost-effective electropolymerized sensor for the detection of food additives in shrimp was developed by George et al., as shown in Figure 9 [131]. 2-aminothiazoleon carbon fiber paper electrode (PAT/CFP) was electro-polymerized in the presence of 4-hexylresorcinol (4-HR) to detect 4-HRin shrimps by the DPV method. This system has a low detection limit of 6.03 nM for 4-HR in shrimps, with the highest recovery rate of 98.23% to 100.14% [131].

Lu et al. developed a loofah-derived biomass carbon-decorated CoFe-CoFe_2_O_4_MIP sensor to detect hazardous chemicals, such as thiamphenicol, in actual milk, honey, and meat samples. Figure 10 gives an overview of the whole fabrication process. The DPV method detected thiamphenicol, and the LOD value was 0.003 µM with reliable recoveries (95.11–105.00%) [132].

Ren et al. designed a MIP-based voltammetric sensor to detect acetaminophen using nitrogen-vacancy graphitized carbon nitride and silver-loaded multi-walled carbon nanotubes (Ag-MWCNTs). The ratio of monomer-template, elution cycle, electro-polymerization cycle, incubation time, and pH was optimized and resulted in linear ranges of 0.007–5 and 5–100 μM with a LOD of 2.33 nM by the DPV method. The recovery ranged from 96.3–100.5% in spiked human urine and serum samples [133]. A tiotropium bromide (TIO)-imprinted electrochemical sensor was developed to detect TIO in pharmaceutical samples by Cetinkaya et al. TIO was analyzed using cyclic voltammetry and DPV detection methods. TIO’s calculated low detection limit is 2.73 fM, with an operating linearity range of 10–100 fM. The recovery rate of real-sample analysis in human serum is 100.77%. They also investigated the stability of the sensor by measuring the recovery rate in a desiccator for 10 days, and the values are as follows: 91.9% on the 3rd day, 89.80% on the 5th day, and 79.19% on the 10th day [134]. In another study, Sulym et al. developed a tetracycline-sensitive electrochemical sensor using L-histidine-MWCNTs-polydimethylsiloxane-5-nanocomposite (L-His-MWCNTs@PDMS). The detection of tetracycline in human serum and tap water samples was determined using CV, DPV, and electrochemical impedance spectroscopy (EIS) techniques. The recovery rate of an experiment performed was found to be 98.92% and 100.60%, with a LOD value of 2.642 × 10^−12^ M [135].

**Table 4 biomimetics-08-00245-t004:** MIP-based electrochemical sensors for environmental applications.

Synthesis Method	Functional Monomer	Detection Method	Analyte	LoD	Recovery Real Sample	Reference
Precipitation polymerization	Vinyl benzene, MAA	DPV	Chloridazon	6.2 × 10^−8^ mol L^−1^	Ground water-95% Surface water-94% Drinking water-96.5% Sea water-92%	[136]
Precipitation polymerization	2-vinylpyridine, AM, MAA	DPAdCSV	Hexazinone	2.6 × 10^−12^ mol L^−1^	River water-95.8%	[137]
Precipitation polymerization	MAA, 2-(5-Bromo-2-pyridylazo)-5-(diethylamino)phenol	DPAdCSV	Uranyl Ions	1.1 × 10^−10^ mol L^−1^	Tap water-99.8% Caspian Sea water-100.4% Persian Gulf water-100.7% River water-99.5%	[138]
	Methylene succinic acid	Potentiometric	Cr (III)	5.9 × 10^−7^ mol L^−1^	River water-98% Sea water-102%	[139]
Precipitation polymerization	MAA	Voltammetric	Para-nitrophenol	3 × 10^−9^ mol L^−1^	Tap water-99.4% River water-100.4%	[140]
	MAA	Square wave voltammetry	Dicloran	9.4 × 10^−10^ mol L−1	Tap water-96.50% River water-100.30% Urine-93%	[141]
Core-shell	MAA	Square wave voltammetry	TNT	0.5 nM	Tap water-(94–100.6%) Sea water-(90–107.5%)	[142]
	MAA, 4-aminothiophenol	CV and DPV	Tetrabromobisphenol-S	0.029 nM	Tap water-(98.7–107.3%)	[143]
Bulk polymerization	2-vinylpyridine	SWV	Diuron	9.0 × 10^−9^ mol L^−1^	River water-(96.1–99.5%)	[144]
Precipitation polymerization	Vinyl pyridine, MAA	SWV	cerium (III)	10 pM	Drinking water-(95–97.3%) Sea water-(102.7–10.4%)	[145]
Bulk polymerization	MAA	SWV	Carbofuran	3 × 10^–10^ M	Tap water-(94–96%) River water-(94–97%) Urine-(91–94%)	[146]
Radical polymerization	MAA	DPV	Diphenylamine	0.1 mM	Synthetic sample	[147]
Electro polymerization		DPV	Cd^2+^	1.62 × 10^−4^ μm	Tap water-(98.5–102.2%) River water-(99.5–100.67%) Milk-(99–109.2%)	[148]
Sol-gel	3-[2-(2-aminoethylamino) ethylamino] propyl-trimethoxysilane	DPASV	Cd^2+^	0.15 μgL^−1^	Tap water and River water- (97.0–101.7%)	[149]
Suspension Polymerization	MAA	DPV	Methylene blue	11.65 µg/mL	-	[150]
Precipitation polymerization	MAA	SWV	Paraoxon	1.0 × 10^−9^ mol L^−1^	Tap water-(101.3%) River water-(103.2%) Lake water-(97.8%)	[151]
Sol gel	3-Aminopropyl triethoxysilane	DPV	Tetrabromobisphenol-A	0.77 nM	Tap water-(96.54–105.78%) Pool water-(92.41–99.27%)	[130]
Precipitation polymerization	MAA	DPV	Pb^2+^	1.3 × 10^−11^ mol L^−1^	Flour-(99.1%) Rice-(103.7%) Tap water-(99.4%) River water-(102.1%)	[152]
Precipitation polymerization	MAA Ethylene glycol dimethacrylate	potentiometric sensor	Cu^2+^	2.0 × 10^−9^ mol L^−1^	Tap water-(101–103%) River water-(100–106%)	[153]
Precipitation polymerization	MAA	DPV	Ag(I)	97 μg L^−1^	Well water-(97.2–98.2%) Aqueduct water-(98.2–103%) Dam water-(97.3–99.6%)	[154]
Precipitation polymerization	MAA	Impedimetric sensor	5-Chloro-2,4-dinitrotoluene	0.1 μM		[155]
Precipitation polymerization	AM	square-wave adsorptive anodic strippingvoltammetry	Methyl green	1.0 × 10^−8^ mol L^−1^	River water-(99.5–103%) Industrial waste water-(93.7–99.3%)	[156]
Electro polymerization	ortho-phenylenediamine	DPV	Acesulfame-K	0.35 µM	Cool drink-(100.8–108%) Candy-(99.6–104%) Tabletop sweetener-(98.4–102.4)	[157]
Electro polymerization	pyrrole	DPV	catechol	0.54 µM	Tap water-(93.90 to 99.69%	[158]
Electro polymerization	l-arginine	Cyclic voltammetry	Tartrazine	0.0027 µM	Soft water-(92.63–105.59%) Orange-flavored jelly powder-(95–100.7%)	[159]
Bulk polymerization	Itaconic acid	DPV	Metribuzin	0.1 pg/mL	Pure samples-(99.29–101.38%) Tomatoes samples-(98.74–102.34%) Potatoes sample-(97.47–103.4%)	[160]
Electro polymerization	*o*-phenylenediamine	DPV	Nitrofurazone	0.18 nmol L^−1^	Milk-(96.06–101.46%)	[161]
Electro polymerization	*MAA*	DPV	ceftriaxone	0.008 µM	Powder-(98.67–101.71%) Urine-(101.44–104.20%)	[162]
	*MAA*	DPV	creatinine	5.9 × 10^−8^ M	Plasma samples-(97.40–119.25%)	[163]
Electro polymerization	o-Phenylenediamine	DPV	Thiabendazole	0.23 μM	Apple-(78.2–86.4%) Pear-(87.7–91.2%) Orange juice-(82.3–87.1)	[164]
Electro polymerization	Pyrrole	DPV	picric acid	1.4 μM	-	[165]
Radical polymerization	MAA	DPV	maleic hydrazide	40 ppb	Onion-(88.5–94.5%) Garlic-(82.2–105.1%) Potato-(80.0–106%)	[166]
Thermal precipitation polymerization	MAA	Voltammetric	2,4-dinitrophenol and 2,4,6 trinitrotoluene	0.59 μM and 0.29 μM		[167]
Electro polymerization	Carbazole	SWV	Nitrobenzene	0.107 μM	Honey-(99–114%)	[168]
Thermal polymerization	MAA; itaconic acid; acrylamide; 2-(trifluoromethyl)-acrylic acid; N, N-Methylene Bis Acrylamide	EIS	Methidathion	5.14 μg/L	Tap water-(98–100.35)	[110]
Precipitation polymerization	MAA	EIS	N-nitrosodimethylamine	0.85 μg/L	Tap water-(99%)	[169]
Self-polymerization	Dopamine	EIS	Dichlorodiphenyltrichloroethane	6 × 10^−12^ mol L^−1^	Raddish-(83–102%)	[170]
Electropolymerization	o-Phenylenediamine	EIS	Alachlor	0.8 nM	Tap water-(95.5–103.5%)	[171]

### 5.2. MIPs-Based Electrochemical Sensors for Bio Applications

The electrochemical biosensor is a self-contained analytical device that recognizes biological elements in direct contact with the electrochemical transduction element to perform the selective and sensitive detection of biological analytes. MIPs-based electrochemical sensors for detecting biological analytes are listed in Table 5. Recently, Buensuceso et al. developed an electro-polymerized poly terthiophene MIP sensor for the detection of dengue, and it was facilitated by the CV method and monitored by electrochemical quartz crystal microbalance (EC-QCM); thus, the spiked buffer solutions of dengue NS1 protein, which has a linear range of 0.2 to 10 μg/mL with a detection limit of 0.056 μg/mL [172]. In another study, the detection of cytochrome C was performed by Campagnol et al., with sub-pico molar level detection in the serum samples and 10^−15^ M in buffer solutions. This MIP system was polymerized by electropolymerization, and the DPV technique was used for electrochemical measurement [173]. Mobed et al. designed a novel genome sensor for *Legionella pneumophila*, a causative agent for Pontiac fever and legionaries’ diseases, using DNA immobilization and hybridization techniques. Thus, the DNA was quantified in a linear range from 1 μM to 1 ZM (Zepto molar). Tang et al. developed a MIP-based electrochemical handheld sensor device for monitoring changes in the cortisol steroid hormone found in various biofluids, including saliva, blood, urine, sweat, and interstitial fluids, and aerometric techniques performed the sensitivity analysis. The detailed functions of this sensor are discussed with the help of Figure 11 [174].

This study uses cyclic voltammetry, square wave voltammetry, and electrochemical characterization of immobilized DNA [174]. Mani et al. developed an L-tryptophan (LTRP) sensor with MIP-assisted silver-decorated silanized graphene oxide. Thus, results with a LOD value of 3.23 × 10^−10^ M and accurate sample analysis to detect LTRP in human blood serum produced a recovery rate of (98–102%) [175]. Charlier et al. performed the electrochemical detection of penicillin G using MIP-based sensors. They investigated the electrochemical characterization through the EIS technique, resulting in a sensing range of 12.5–100 ppb [176]. To detect serotonin, Tertis et al. developed MIPs containing chitosan and graphene oxide-based novel electrochemical sensors. The LOD of serotonin detection was 1.6 nM. The actual sample analysis was performed in human serum (93.0–95.8%), artificial tears (98–102%), and artificial saliva (97–110%). The DPV technique is used to investigate the real sample analysis [177]. Diouf et al. developed a MIP-based nonenzymatic electrochemical glucose sensor for measuring the glucose contents in saliva and finger prick blood samples. Various electrochemical techniques, such as DPV, EIS, and CV, performed glucose detection. The operating range of the MIP sensor was from 0.5 to 50 μg/mL, with an excellent detection limit of 0.59 μg/mL. Thus, satisfactory results (R^2^ = 0.99) were obtained for real saliva glucose determination compared with a finger prick blood sample (Figure 12) [178].

Oliveira et al. designed a disposable electro-polymerized MIP sensor to detect carbohydrate antigen (CA 15-3), a breast cancer biomarker. *Ortho*-phenylenediamine (oPD) was used as a functional monomer for constructing MIP polymeric films. The system has a LOD of 1.16 UmL^−1^ with a linear range of detection of 5–35 UmL^−1^ and an actual sample recovery rate of 101.8–104.3%, tested in human serum samples [179]. Another work on levodopa (a precursor to dopamine) detection in biofluids was investigated by Pourhajghanbar et al. This system showed a low LOD value of 10 nmol L^−1^. Electropolymerization was used to fabricate the MIP with levodopa as a template and dopamine + resorcinol as a bifunctional monomer. The actual sample was analyzed by square wave voltammetric techniques with recovery rates in blood serum real samples (93.05–107.43%) and plasma (93.99–107.6%) [180].

**Table 5 biomimetics-08-00245-t005:** MIPs-based electrochemical sensors in biomedical applications.

Synthesis Method	Functional Monomer	Detection Method	Analyte	LOD	Recovery Real Sample	Reference
Electro polymerization	3-aminophenol	Amperometry	Tau-441 protein	0.01 pmol/L		[181]
Electro polymerization	Methylene blue	DPV	Lysosome	4.26 fM	Serum-(94–108%) Urea-(98–109%)	[182]
Electro polymerization		DPV	Immunoglobulin G	0.017 ngmL^−1^	Serum-(97.36–100.98%)	[110]
Electro polymerization	Aniline	CV and EIS	Histamine	1.07 nM	-	[183]
Electro polymerization	polyacrylamide	DPV	Dopamine andadenine	0.12–0.37 μM and 0.15–0.37 μM	Serum-dopamine (96–108%) Serum-adenine (92–104%)	[164]
Chemical polymerization	Aniline	EIS	Tryptophan	8 pM	Milk-(98.4–101.4%)	[184]
Electro polymerization		DPV	Cortisol	20.2 pM		[185]
Electro polymerization	poly o-phenylenediamine	CV, EIS, and SWV	Cortisol	200 fM	Saliva-(91–105%)	[186]
Electro polymerization		CV and EIS	Aflatoxin B1	12.0 pg L^−1^	Milk-(97–104%)	[187]
Photopolymerization	MAA	DPV	Cholesterol and cholestanol	0.01 μM	Serum-(93.6–101.03%)	[188]
Photopolymerization	AM	EIS	Neutrophil gelatinase-associated lipocalin (NGAL)	0.07μg/mL	Real NGAL-91%	[189]
Electro polymerization		EIS	SARS-CoV-2	10 to 10^8^ PFU/mL	Saliva-(98 to 104%)	[190]
Free radical polymerization	vinyl phosphonic acid		sarcosine	0.04 µM		[191]
One pot method		DPV	Creatinine	2 × 10^−2^ pg/mL	Serum, urine (93.7–109.2%)	[192]
	Methyl methacrylate	DPV	H. pylori	0.05 ng mL^−1^	Blood-96%	[193]
Electro polymerization	2-aminophenol	EIS	Galectin-3	30 ng/mL		[194]
Electro polymerization	Dopamine	DPV	Trypsin	0.75 pg/mL	Urine-(94–100.2%) Serum-(98.4–114%)	[195]
Precipitation polymerization	Methyl methacrylate	DPV	serum amyloid A	0.01 pM		[196]
Electro polymerization	Methyl methacrylate	EIS	Follicle-stimulating hormone (FSH)	0.1 pM	Blood samples (90–98.79)	[197]
Electro polymerization	Eriochrome Black T	EIS	Interleukin-1β	1.5 pM		[198]
Co-Electropolymerization	carboxylated pyrrole	EIS	Interleukin-6 (IL-6)	0.02 pg/mL		[199]
Bulk polymerization	Methyl methacrylate	CV and EIS	Anandamide	0.01 nM	Blood samples-(93.48 and 90.08%)	[200]
Electro polymerization	3-aminophenylboronic acid	DPV	Lactate	0.22 μM	Sugarcane vinasse-(97.7 to 104.8%)	[109]
Electro polymerization	3-aminophenylboronic acid	CV and EIS	Interleukin-6	1 pg/mL		[179]

From these studies, MIPs-based sensors could provide many advantages, such as cost-effectiveness, superior stability, rapid, easy synthesis, selectivity, and high sensitivity, which can be utilized for biomedical applications. Apart from all these advantages, one of the main limitations of MIPs is the hydrophobic or hydrophilic nature of the monomer, which influences polymer imprinting. Futuristic advancements in MIP-based technologies could resolve this problem through artificial intelligence [198,199]. Unquestionably, massive investigations are still needed to improve the selectivity and potential of shape recognition in sensors based on MIPs.

## 6. Conclusions and Future Perspectives

MIP-based sensors have been found to have enhanced specificity and sensitivity. However, several challenges need to be addressed before the technology can be commercialized. Some of the challenges are listed below, which require thorough investigations so that MIPs can reach the market soon. The challenges include: (a) MIPs perform best under invitro lab conditions. However, they are found not to serve as expected in real-world samples. Thus, more research is needed to enhance their sensitivity, specificity, and, finally, reproducibility in real-world samples. (b) MIPs are found to have high specificity for single analyte detection. However, multiple sensors are currently much preferred. MIP-based multiplex sensors that can detect several analytes simultaneously are to be developed. (c) There is a strong need for innovations in the materials and manufacturing aspects when MIPs are combined with nanomaterials. There is a need for cheaper, more reliable, and scalable fabrication technologies. A few different types of functional monomers and cross-linking agents are available to synthesize MIPs. The chemical reagents used face high capital costs and low conversion efficiency, making it challenging to transition from laboratory to factory mass production and unable to maximize commercial conversion. For example, in-situelectropolymerization is one of the processes involved in MIP synthesis. They are expensive to operate and need other, cheaper strategies for the same. (d) There is a need to increase the speed of the test procedures in the case of a point-of-care (PoC) setup [201]. (e) MIPs typically function best in hydrophobic organic solvents; however, in the future, this could obstruct the formation of pre-polymerization complexes and interfere with the interaction between the template and the monomer. Therefore, hydrophilic polymers are needed. (f) Template leakage is another issue that frequently plagues MIPs, leading to the formation of molecularly imprinted materials with asymmetrical particle sizes, non-uniform recognition sites, and poor affinities. (g) Need for biodegradable and environmentally friendly biopolymer-based MIPs. Only recently have there been a few reports on biopolymer-based MIPs. Some of the explored biodegradable polymers are silk [202] and chitosan [203]. More research is needed to develop more of these bio-MIPs. Researchers across the globe are currently searching for solutions to these issues. In the near future, hand-held sensors based on MIPs are expected to be developed, allowing users to detect any analyte in a PoC setup. We hope these sensors could completely transform the healthcare sector by lowering the cost of sensors and enhancing clinical outcomes. Thus, MIPs combined with several nanomaterials could be developed as a PoC/PoU sensing platform for efficiently detecting biomarkers and other contaminants with high reproducibility, specificity, and sensitivity.

## Figures and Tables

**Figure 1 biomimetics-08-00245-f001:**
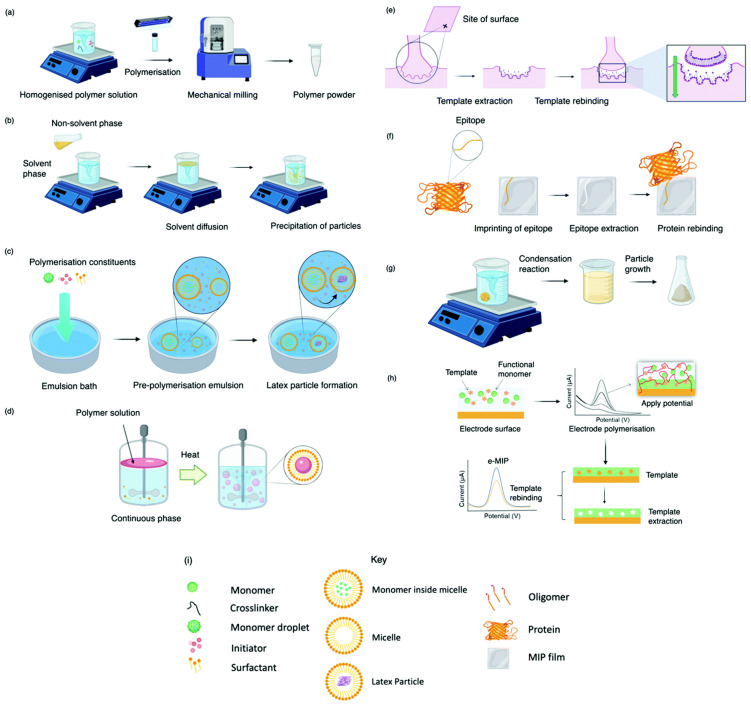
Schematic representation depicting various polymerization techniques: (**a**) bulk, (**b**) precipitation, (**c**) emulsion, (**d**) suspension, (**e**) surface, (**f**) epitope, (**g**) sol-gel, (**h**) electro-polymerization, and (**i**) each technique’s related key [25].

**Figure 2 biomimetics-08-00245-f002:**
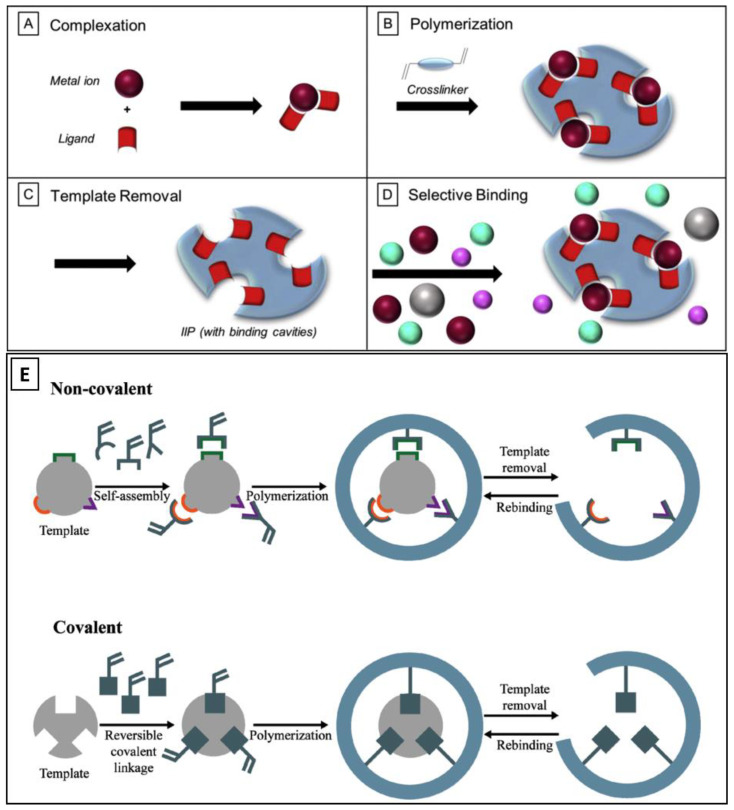
Illustration of three prominent imprinting techniques for MIPs. (**A**–**D**) depicts the steps involved in the metal ion-mediated MIP synthesis; (**E**) depicts the covalent and non-covalent imprinting techniques. The figure is reproduced with permission from Elsevier [62,63].

**Figure 3 biomimetics-08-00245-f003:**
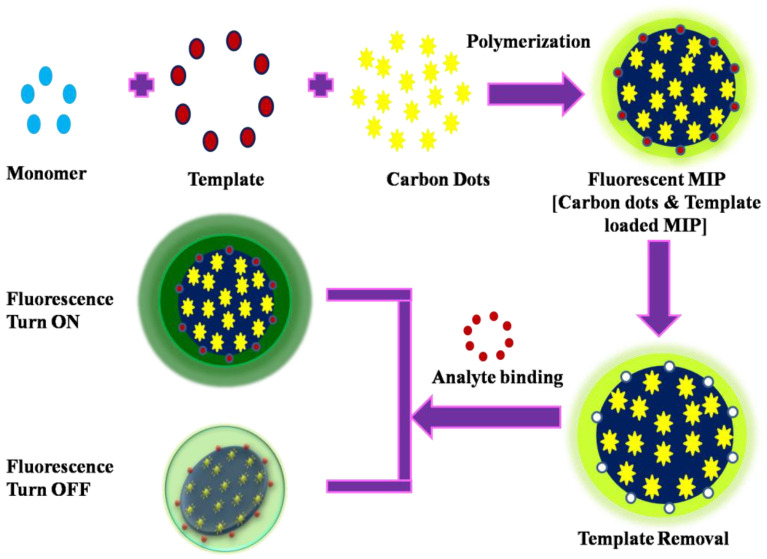
Schematic illustration of the working principle of MIPs-based optical sensors.

**Figure 4 biomimetics-08-00245-f004:**
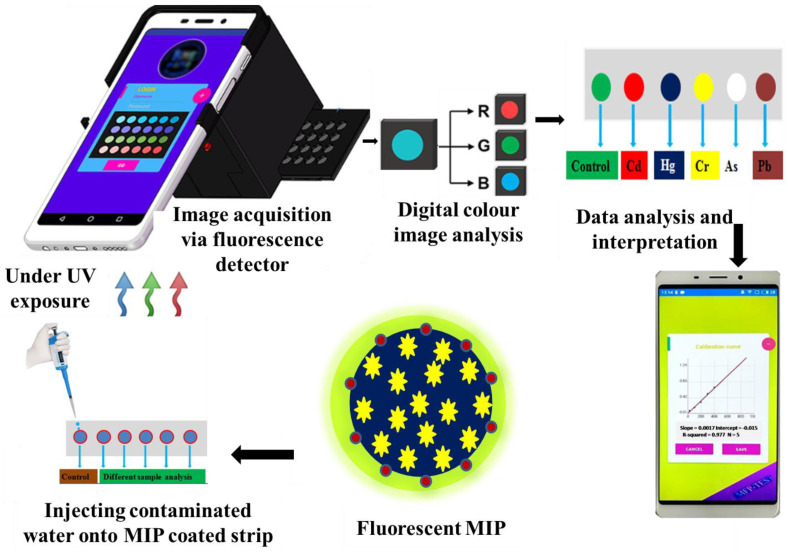
Schematic illustration of the fabrication of MIPs-based optical sensors.

**Figure 5 biomimetics-08-00245-f005:**
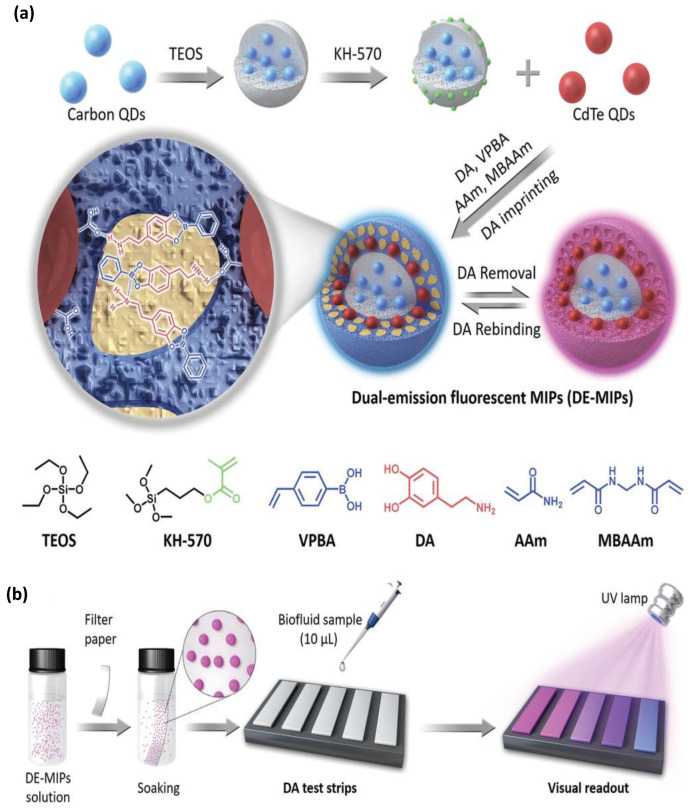
(**a**) Schematic representation of dual-emission fluorescent MIP nanoparticles (Dual emission-MIPs) with specific dopamine affinity. (**b**) The dual emission-MIPs-coated filter paper as a facile dopamine test strip for visual detection. Copyrights reproduced with permission from John Wiley and Sons [113].

**Figure 6 biomimetics-08-00245-f006:**
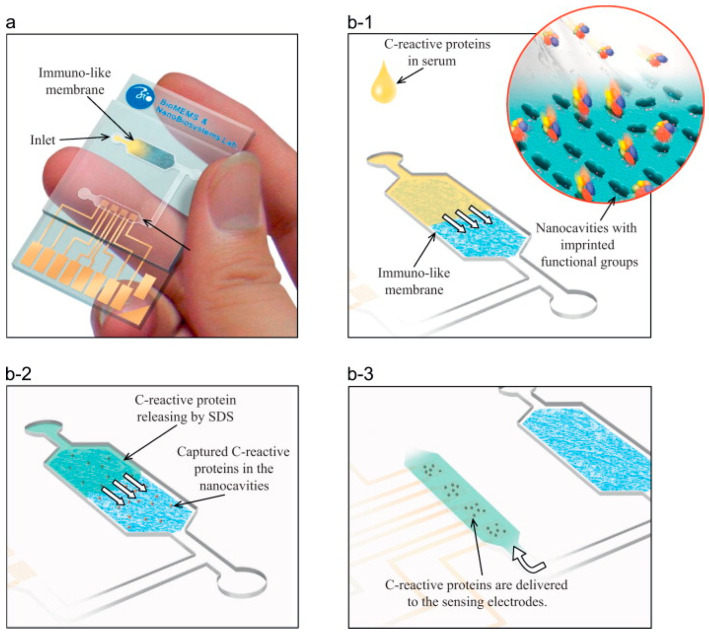
The representation of the MIP-based nanocavities (immuno-like membrane in the microfluidic system (**a**); loading of serum samples into the microfluidic system and capturing C-reactive protein from serum samples (**b-1**); loading of SDS and releasing of C-reactive protein from the immuno-like membrane (**b-2**); delivery of SDS with C-reactive protein to the electrodes (**b-3**)). The figure is reproduced with permission from Elsevier [120].

**Figure 7 biomimetics-08-00245-f007:**
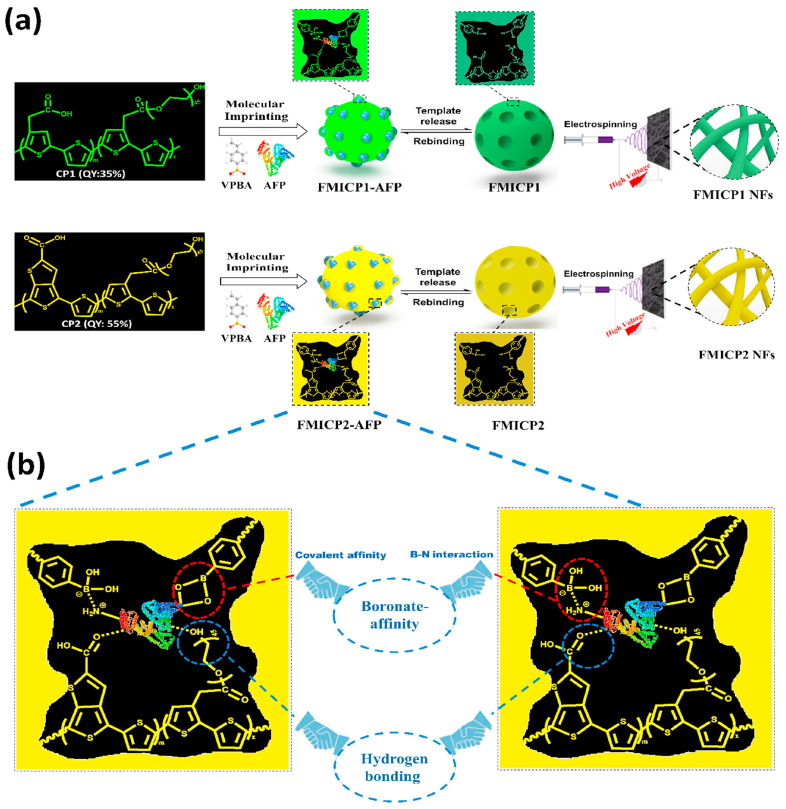
Principles and strategies of FMICP and FMICP NF biomarker assays: (**a**) Synthesis of the conjugated polythiophenes linked—molecular-imprinting strategy and fabrication of their fluorescent nanofibers using an easy and low-cost electrospinning approach, as well as their interactions with the AFP (alpha-fetoprotein) biomarker. (**b**) Mechanism of dual-emission CPs linked with boronate-affinity molecular-imprinting strategies The figure is reproduced with permission from Elsevier [121].

**Figure 8 biomimetics-08-00245-f008:**
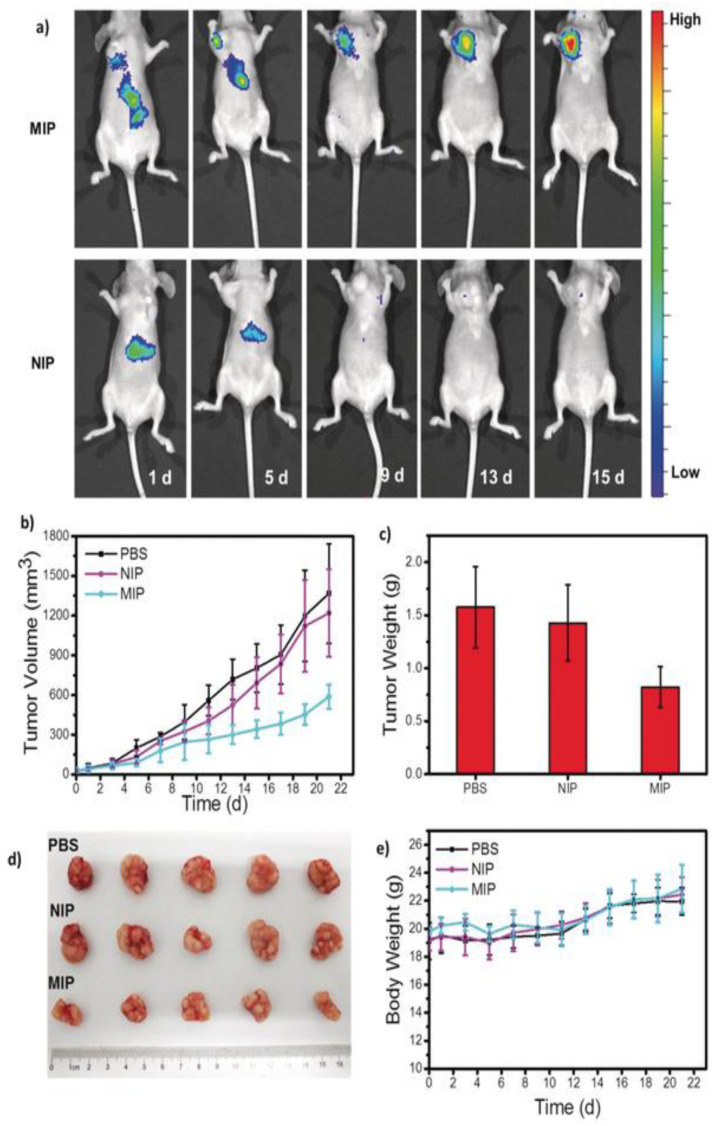
(**a**) Schematics showing the mechanism of inhibition of breast cancer growth by MIP NPs; (**b**) Confocal images of breast cancer cells stained with FITC-doped silica NPs imprinted with HER2-glycan (MIP) and nonimprinted NPs showing absence of fluorescence; (**c**) In vivo imaging of the tumor after intravenous injection of MIP and NIP doped with an infrared dye confirms the accumulation of MIP at the tumor site; (**d**,**e**) Effect of MIP/NIP on the tumor volume in mice after treatment. The figure is reproduced with permission from John Wiley and Sons [126].

**Figure 9 biomimetics-08-00245-f009:**
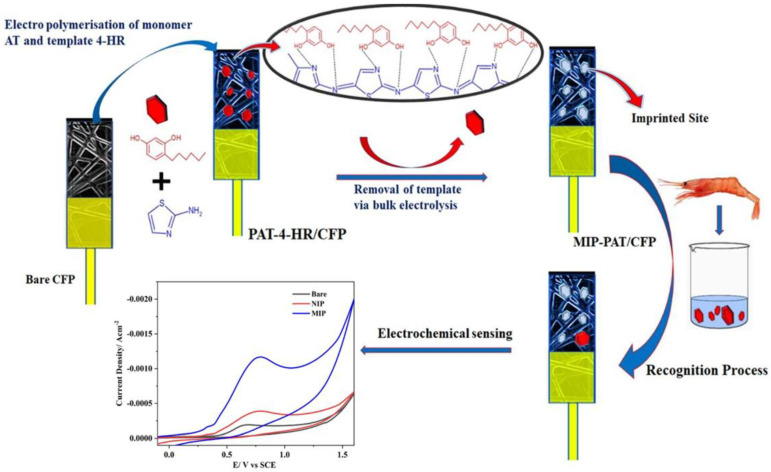
Schematic illustration of MIP-PAT/CFP preparation and its sensing process. The figure is reproduced with permission from Elsevier [131].

**Figure 10 biomimetics-08-00245-f010:**
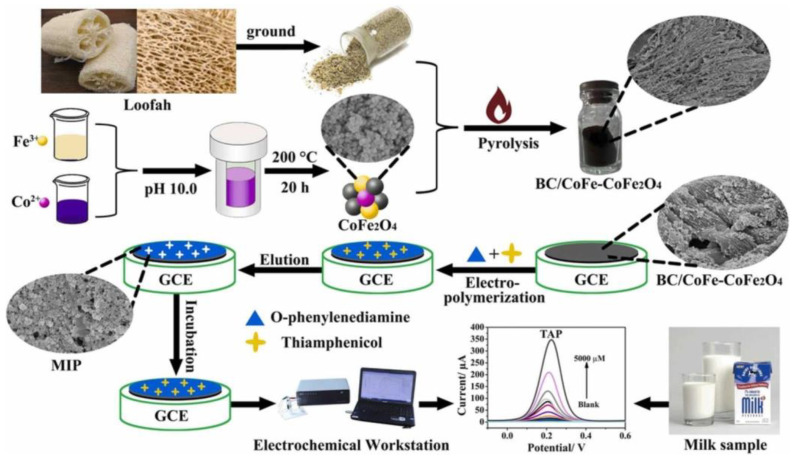
Schematic figure of the MIP/BC/CoFe-CoFe_2_O_4_ nanocomposite preparation process and electrochemical DPV response of thiamphenicol in milk samples. The figure is reproduced with permission from Elsevier [132].

**Figure 11 biomimetics-08-00245-f011:**
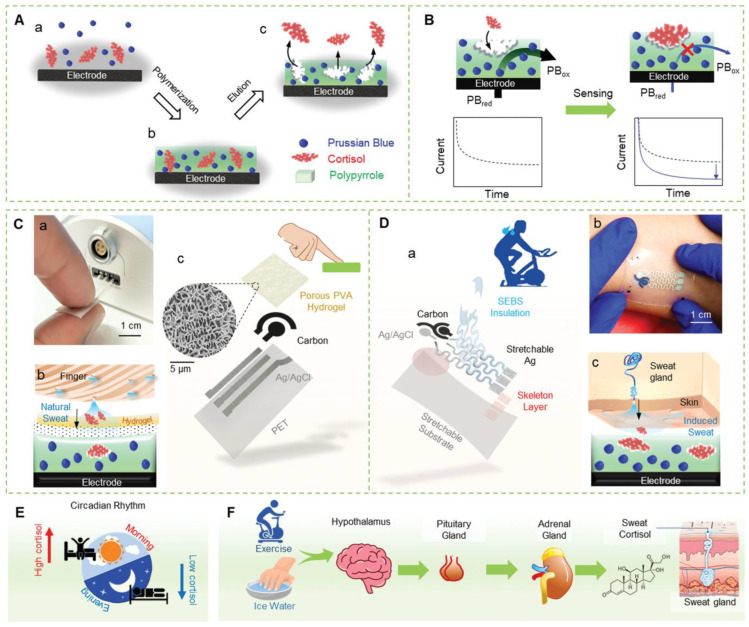
Schematic overview of a MIP-based stressless cortisol sensor. (**A**) Fabrication of the MIP layer; (**B**) cortisol-entrapped template eluted from the polymerized polypyrrole, corresponding MIP layer after the cortisol elution, where the cortisol-specific cavities are formed in the electrodes; (**C**)Images of touch-based fingertip cortisol sensors; (**D**) the stretchable epidermic cortisol path; (**E**) the illustration of the circadian rhythm; and (**F**) the description of cortisol secretion by physical movements. The figure is reproduced with permission from John Wiley and Sons [174].

**Figure 12 biomimetics-08-00245-f012:**
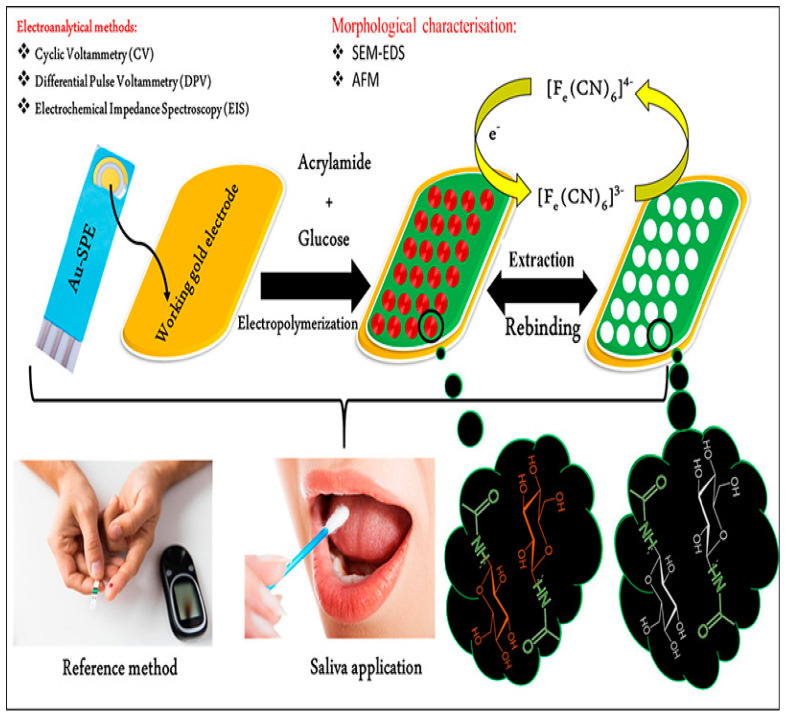
Schematic representation of fabrication of MIP-based nonenzymatic electrochemical glucose sensor. The figure is reproduced with permission from Elsevier [178].

**Table 1 biomimetics-08-00245-t001:** Lists of most commonly used functional monomers, initiators, crosslinkers, and porogens for the synthesis of MIPs.

S. No	Functional Monomer	Crosslinker	Initiator	Porogenic Solvents	Morphology	Polymerization Type	Reference
1	4-Vinyl Pyridine, Methacrylic Acid (MAA), Itaconic Acid. N-Vinylimidazole, Allylthiourea, Acrylamide, N-Methacryloyl-(L)-Cysteine, 2-VinylPyridine	Ethylene glycol dimethacrylate (EGDMA), Divinyl Benzene	AIBN, 2-Hydroxyethyl Methacrylate, Lauryl Peroxide, and Benzoyl Peroxide	Acetone, Cyclohexanol	Monolith	Bulk polymerization	[26]
2	Dithizone, N-[3-(2-Aminoethylamino) Propyl] Trimethoxysilane, 3-Isocyanatopropyl Triethoxysilane	Tetraethoxysilane	Ammonia	-	Dendritic	Sol-gel process	[27]
3	N-Propylacryl Amide	N,N-Methylene-Bis-Acrylamide (Mbam)	Ammonium Sulfate		Microspheres	Surface grafting polymerization	[28]
4	Chitosan	Epichlorohydrin			Microspheres	Suspension Polymerization	[29]
5	Acrylamide and β-Cyclodextrin	Epichlorohydrin	Ammonium persulfate			Emulsion polymerization	[30]
6	N-Methacryloyl-(L)-Cysteine	Metylenebis (Acrylamide)	Ammonium Persulfate		Membranes	Multi-step swelling polymerization	[31]
7	2-Methacryloylamido Histidine	Poly(Ethylene Glycol) Diacrylate	Ammonium Persulfate		Membranes	Multi-step swelling polymerization	[32]
8	MAA, Divinyl benzene	EGDMA			Micro particles	Precipitation polymerization	[33]
9	O-amino phenol				Nanoparticles	Electro deposition	[34]
10	O-Phenylene diamine				Nanowires	Electro deposition	[35]

**Table 2 biomimetics-08-00245-t002:** Advantages and limitations of the polymerization techniques used for the synthesis of MIPs.

S. No.	Polymerization Type	Advantages	Limitation	References
1	Bulk	Cost-effective method. Ease in preparation. Better control over the size of MIP particles synthesized	Low selectivity and reproducibility. Use of time-consuming processes. Need an ample amount of eluent to remove the template. No control over the shape of MIPs generated. The MIP obtained requires grinding, which results in some irregularities in the shape of the particles. Requirement of the huge amount of porogens during the fabrication process.	[37,58]
2	Suspension	Spherical particles with high porosity are obtained by this method.	Due to the influence of the dispersing media, MIPs produced in this manner have poor recognition sites compared to other techniques. This method is suitable only for hydrophobic monomers and initiators.	[43,59,60]
3	Emulsion	Spherical MIPs are formed. The binding sites on the surface of the spherical MIPs are distributed evenly, and the reuse rate is high for MIPs.	Due to their strong polarity and hydrogen bond-forming capacity, the water molecules in the aqueous phase affect the interaction between the template and monomer, resulting in an impaired imprinting process. This polymerization technique’s precipitation and separation processes are complicated as they require demulsifiers and coagulants, which are challenging to purify in the end. These impurities affect the physical properties of the MIPs formed.	[38,40]
4	Precipitation	This process results in high-purity MIPs compared to synthetic approaches like emulsion and suspension polymerizations. Regular-shaped MIP beads are obtained in good yields. Easy and less time-consuming method.	The precipitation only occurs when the polymeric chains are large enough to be insoluble in the reaction mixture. There is a need for high-speed homogenization to form particles of uniform size. The particles formed in the reaction are affected by slight variations in several factors, including the polarity of the solvent, the reaction temperature, and the stirring rate. Thus, the reaction conditions are to be monitored efficiently.	[41,42]
5	Multi-step swelling	This method results in uniform and monodispersed spherical MIP particle.	This method requires sophisticated procedures that are time-consuming. More importantly, the swelling degree of the MIPs should be cautiously controlled. The swelling can negatively influence the recognition ability of the MIPs. Thus, the swelling property of MIPs needs to be thoroughly evaluated to avoid losing its memory effect.	[51,60]
6	Surface imprinting	The mass transfer rate and efficiency are increased because of the increasedexposure of recognition sites on the surface. This results in better adsorption and specific recognition capacity, making it more suitable for separation or sensing applications. The amount of eluent needed for removing the template is meager compared to other bulk techniques.	The surface imprinting process is complicated, with many process parameters involved in obtaining a uniform MIP film. Thus, this is a time-consuming and expensive process.	[46]
7	Electrochemical	Deposition of MIPs with a precise thickness on an electrode surface is possible. There is little or no requirement for eluents to remove the template molecules. Crosslinkers or initiators are not required.	This is an expensive polymerization technique. The optimization of the MIP coating process is a complicated and time-consuming process. For instance, a thin coating results in very few recognition and rebinding sites. On the other hand, the removal of templates becomes complex, resulting in poor rebinding of analytes in the case of thicker coatings.	[61]

**Table 3 biomimetics-08-00245-t003:** MIPs in different environmental applications.

Optical Sensor Material	The Physical Form of Sensors	Detection Method	Monomer	Target	Sample	LoD	Reference
MIP	Paper	UV-Visible	MAA + Polyethyleneimine	Cd (II)	Lake water	1–100 ng/mL	[83]
MIP-C-dots	Film	Fluorescence	acrylic acid (AA) + methylacrylate (MA)	2,4- dinitrotoluene	Lake and tap water	1–15 ppm, 0.28 ppm	[84]
MIP-C-dots	Film	Fluorescence	APTES	Cetricine	Urine, Saliva	0.5–500 ng/mL, 0.41 ng/mL	[85]
Silanizedmagneticgraphene-MIP	Capillary tube	Chemiluminescence	Acrylamide (AM)	Dopamine	Urine, dopaminehydrochloride injection	8–200 ng/mL, 1.5 ng/mL	[86]
MIP/Chromatographypaper	Paper disk	Chemiluminescence	AM	2,4-dichlorophenoxyaceticacid	Lake and tap water	5 pM–10 μM, 1 pM	[87]
MIP-Magnetic NP	Nanoparticles	Chemiluminescence	MAA	Dibutyl phthalate	Juice	3.84 × 10^−8^–2.08 × 10^−5^ M	[88]
MIP	Optical fiber	Surface Plasmon resonance	MAA	Furfural	Transformer oil	9–30 ppb	[89]
MIP	Optical fiber	Surface Plasmon resonance	MAA	Profenofos	PBS	2.5 × 10^−6^ μg/L	[90]
MIP	Nanoparticles	Surface Plasmon resonance	N-methacryloyl-(L)-histidinemethyl ester	Histamine	Cheese	0.58 ng/L	[91]
MIP	Nanofilm	Surface Plasmon resonance	N-methacryloyl-(L)-tryptophan methylester	Carbofuran, dimethoate	River water	7.11 (carbofuran); 8.37(dimethoate) ng/L	[92]
MIP-Ag NP	Film	Surface Plasmon resonance	N-methacryloyl-(L)-histidinemethyl ester	Escherichia coli	Urine	15–1,500,000 CFU/mL	[93]
MIP	Nanoparticles	Raman scattering	MAA	Propranolol	Human Urine	7.7 × 10^−4^ M	[94]
MIP-Au NP	Core-shell	Raman scattering	3-(triethoxysilyl)propylisocyanate (TEPIC)	Bisphenol A	Surface water, plastic-bottled beverages	2.2 × 10^−6^–10^−4^ M, 5.37 × 10^−7^ M	[95]
MIP-Ag	Core-shell	Raman scattering	AM	Glibenclamide	Water	1 ng/mL–100 μg/mL	[96]
Au-MIP	Nanoparticles	Raman scattering	MAA, AM	2,6-dichlorophenol	Water	0.02 nM	[97]
MIP-Au NP	Fine particles	Raman scattering	MAA	Atrazine	Apple Juice	0.0012 (SERS) mg/L	[98]
Magnetic MIP	Nanoparticles	Fluorescence, Raman scattering	Poly(ethylene-co-vinylalcohol)	Phenylalanine	Human urine	7–100 (F); 5–800 μg/mL (RS)	[99]
MIP	Membrane	UV-Visible	Itaconic acid	Phenol	Drinking, natural, and wastewater	50 nM–10 mM, 50 nM	[100]
MIP	Fine particles	Raman scattering	MAA	Melamine	Milk	0.005–0.05 mM, 0.012 mM	[101]
MIP-Magnetic NP	Core-shell	Raman scattering	MAA	Ciprofloxacin	Fetal bovine serum	10^−7^–10^−4^ M	[102]
MIP	Film	Surface Plasmon resonance	MAA	Histamine	Fish	25 μg/L	[103]
MIP	Optical Fiber	Surface Plasmon resonance	MAA	L-nicotine	Ultrapure water	1.86 × 10^−4^–10^−3^ M	[104]
MIP-QD	Nanocomposite	Fluorescence	APTES	Thiamphenicol	Urine	0.04 μM	[105]
MIP-CdSeS/ZnSQD	Glass slide	Fluorescence	MAA	Sulfasalazine	Human plasma and urine	0.02–1.5 μM, 0.0071 μM	[106]
MIP-QD	Composite	Fluorescence	APTES	Tetrabromobisphenol-A	Electronic waste	1–60 ng/mL	[107]
MIP	Au Nanocomposite	Fluorescence	APTES	Bisphenol A	Seawater	0.1–13 μM	[108]
MIP	Hollow Nanoparticles	Fluorescence	Acrylamide	λ-cyhalothrin	Canal water	10.26–160 nM	[16]
CdTe QD-MIP	Composite	Fluorescence	Acrylamide	λ-cyhalothrin	River water	0.1–16 μM	[109]
MIP	Colloidal array		MAA	Hexanitrohexaaziasowurtzitane; Hexahyro-1,3,5-triazine; 2,4,5-trinitro toluene; 2,4-dinitrotoluene; 2,6 dinitritolune; 1,3,5-trinitrobenzene			[110]

## Data Availability

All the data and materials that support the results or analyses presented in their paper are freely available upon request.

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
