# Peer review of "Molecularly Imprinted Polymer-Based Biomimetic Systems for Sensing Environmental Contaminants, Biomarkers, and Bioimaging Applications"

_biomimetics, 2023, doi:10.3390/biomimetics8020245_

Round 1

Reviewer 1 Report

When I examine the entire article, I regret to say that the text has many inconsistencies. It was made sloppy in the text's writing and the presentation of the scientific data. Some are given below and highlighted in a yellow pdf document.

If the entire article is not edited, I think it should be rejected.

There are errors in punctuation throughout the article. There should be a space before the references given in the text. (Example: 2,3,4 15.6. etc.) Please check and edit all. A space should be left before the other sentence after the dot mark is placed at the end of the sentence. (Page 3, line 108, 110 etc) Please check and edit all the text.

Page 4, line 115 “related key.[35.]” revise as given related key [35].

Page 5, line 146 “10-100nm” revise as given 10-100 nm (Give space between number and unit). Similar errors should be checked in all the manuscripts.

Page 5, line 153 ([45–48]) and  161 ([45–47]) given same references. Is it possible? 2 different polymerisation techniques were given, and the same references were cited.

To make a general assessment of Title 2. Preparation methods of MIPs, seems that it was prepared rather carelessly. Information about the advantages and disadvantages of the techniques used could be given. There are excellent groups study on this subject. Authors should be encouraged to rewrite this section with these considerations in mind.

The resolution of the figures is very low and not possible to read. Please supply the high resolution. 

Author Response

Response to reviewers’ comments

Molecularly imprinted polymers (MIPs)-based biomimetic systems for sensing environmental contaminants, biomarkers and bioimaging applications

KalaipriyaRamajayama,d, SelvaganapathyGanesana,d, PurnimajayasreeRameshb,d, Maya Beenab,d, KokulnathanThangaveluc* and ArunkumarPalaniappand*

aDepartment of Chemistry, School of Advanced Sciences (SAS), Vellore Institute of Technology (VIT), Vellore 632014, Tamil Nadu, India.

bSchoolof Biosciences and technology (SBST), Vellore Institute of Technology (VIT), Vellore 632014, Tamil Nadu, India

cDepartment of Electro-Optical Engineering, National Taipei University of Technology, Taiwan

dCentre for Biomaterials, Cellular and Molecular Theranostics (CBCMT), Vellore Institute of Technology (VIT), Vellore 632014, Tamil Nadu, India.

The authors would like to thank the reviewers for their comments, suggestions, and comprehensive evaluation of our manuscript. The critical comments and suggestions encourage the authors to improve the quality of the manuscript. We also believe that the manuscript is now more concise, clear, and relevant to the readers. In the revised manuscript, the changes are highlighted in yellow colour. Here are the point-by-point responses to the reviewer’s comments.

Reviewer 1

  1. There are errors in punctuation throughout the article. There should be a space before the references given in the text. (Example: 2,3,4 15.6. etc.) Please check and edit all. A space should be left before the other sentence after the dot mark is placed at the end of the sentence. (Page 3, line 108, 110 etc) Please check and edit all the text.

As per the reviewer’s suggestion, the whole text and the errors in the manuscript are corrected and verified.

  1. Page 4, line 115 “related key.[35.]” revise as given related key [35].

As per the reviewer’s comments, the sentence is modified with appropriate space and punctuations. Please see line 112.

  1. Page 5, line 146 “10-100nm” revise as given 10-100 nm (Give space between number and unit). Similar errors should be checked in all the manuscripts.

As per the reviewer’s comments, space is included between number and the unit. Please see line number 140. Similar errors are identified and corrected throughout the manuscript.

4.Page 5, line 153 ([45–48]) and 161 ([45–47]) given same references. Is it possible? 2 different polymerisation techniques were given, and the same references were cited.

The correct references for the respective polymerization techniques have been updated. Please see line 140 and 144.

  1. To make a general assessment of Title 2. Preparation methods of MIPs, seems that it was prepared rather carelessly. Information about the advantages and disadvantages of the techniques used could be given. There are excellent groups study on this subject. Authors should be encouraged to rewrite this section with these considerations in mind.

We have modified this section as per the reviewer’s comments. We have now included the information about the advantages and limitations of each technique in table 2.

  1. The resolution of the figures is very low and not possible to read. Please supply the high resolution. 

As per the reviewer’s suggestion, we have provided high resolution figures.

Reviewer 2 Report

1. Esp. in abstract. Spell it out

2. Table 1 is incomplete. The preparation of NanoMIPs (bottom up approach) needs to be included; 

3. Fig. 1h: DPV data image: this should be CV image for electropolymerization of MIP

4. Line 120 replace grinded with ground

5. Line 132: what is suspending agent?

6. lines 184-185: needs expanding; it is vague

7. Fig. 3. should have a better size scale. eg. size of monomer is not relative to size of MIP produced. It can be misleading

8. The manuscript uses "cost-effective" 4 times; this needs qualifying e.g. can a cost-benefit analysis be done to show cost-effectiveness rather than just stating it?

9. lines 564-566: repetitive circular argument; rephrase

10.  Line 520: what do you mean by significant monomer? do you mean functionsal monomer; use generally accepted terminology throughout.

11. Section 2.7: why focus on just conducting polymers?? what about the work done with non-conducting polymers?

12. Line 382 contradicts line 565. On the one-hand you say MIPs are biocompatible and on the other you say MIPs are not biocompatible. Needs clarification.

13. Table 3: Since 2106, there has been a growing body of work around using EIS for electrochemical analysis of MIPs as opposed to/in addition to CV and DPV. This also needs to be captured here.

Generally good prose throughout.

Author Response

Reviewer 2

  1. Esp. in abstract. Spell it out

We have modified the word as “especially” in the abstract. Please see in line # 19.

  1. Table 1 is incomplete. The preparation of NanoMIPs (bottom up approach) needs to be included

We have included the bottom up methods as well in the table 1 as per the reviewer’s comments and is highlighted in the table.

  1. Fig. 1h: DPV data image: this should be CV image for electropolymerization of MIP

We thank the reviewer for pointing this out. However, the image was taken from a published review article with permission and we don’t have rights to change it further. However, there are also few reports on the DPV-based electro-polymerization methods. We have cited few of them in the manuscript in section 2.7, line number 176-178.

  1. Line 120 replace grinded with ground

We have modified the same as per the reviewer’s comments. Please see in line # 119.  

  1. Line 132: what is suspending agent?

The suspending agents are substances added in colloidal systems to prevent aggregation of particles and thus keep them suspended longer in the continuous phase. They are typically used for enhancing stability of the colloidal system. Please see line #126-128.

  1. Lines 184-185: needs expanding; it is vague

We have modified the sentences as per the reviewer’s suggestion. Please see line # 164-193.

  1. Fig. 3. should have a better size scale. eg. size of monomer is not relative to size of MIP produced. It can be misleading

We thank the reviewer for pointing out this very important point. As per the reviewer’s suggestion, size of the monomers (blue spheres) has been changed and updated in figure 3.

  1. The manuscript uses "cost-effective" 4 times; this needs qualifying e.g. can a cost-benefit analysis be done to show cost-effectiveness rather than just stating it?

There are several reports in the literature where the cost-effectiveness of MIPs when compared to antibodies are highlighted. They are also cited in our manuscript [38-39,166]. Performing a cost-benefit analysis will be out of scope of this review paper. However, we shall keep this in mind and perform a cost-benefit analysis for our MIP-based future research works.

  1. Lines 564-566: repetitive circular argument; rephrase

The sentences in the above lines were rephrased. Please see line # 569-572.

  1. Line 520: what do you mean by significant monomer? do you mean functional monomer; use generally accepted terminology throughout.

We have removed the word “significant” and changed into “functional”. Please see line # 527.

  1. Section 2.7: why focus on just conducting polymers?? what about the work done with non-conducting polymers?

We thank the reviewer for pointing this out. We agree with the reviewer that both conducting and non-conducting polymers could be synthesized using electro-polymerization method. We have included both of them now in the manuscript. Please see section 2.7 as well as in the table 4 and 5. However, electrochemical sensors use conducting MIPs more often than their non-conducting counterparts.

Also, the electrochemical polymerization necessitates monomers to be electrochemically active. The polymerization process is initiated by the production of free radicals, upon electrochemical oxidation of the monomers. Thus, the monomer should be electrochemically active.

  1. Line 382 contradicts line 565. On the one-hand you say MIPs are biocompatible and on the other you say MIPs are not biocompatible. Needs clarification.

We thank the reviewer for pointing this contradicting lines. Actually, MIPs are reported to be biocompatible and less immunogenic. However, they are not biodegradable, which is a major limitation. To overcome this limitation, MIPs are recently made using few biodegradable polymers like silk, chitosan to name a few. We have modified the same in the manuscript. Please see line 569-572 where we have modified.

  1. Table 3: Since 2106, there has been a growing body of work around using EIS for electrochemical analysis of MIPs as opposed to/in addition to CV and DPV. This also needs to be captured here.

We have also included papers on EIS-based electrochemical analysis in table 3 as per the reviewer’s comments and highlighted in yellow color.

Reviewer 3 Report

The review by Palaniappan et al. is an important paper on the generalization of methods for the synthesis and applications of polymers with molecular imprints. However, before it can be published, a number of serious shortcomings must be eliminated:

1. Figures in the text should be in the manuscript after they are mentioned in the text and not before the sections in which they are described.

2. On page 5, section 2.5 breaks off in the middle of a sentence. References 50-52 were missed.

3. On page 7, reference 78 is mentioned before the corresponding references 75-77.

4. There are no references to Tables 2-4 in the text. In addition, relevant comments regarding the data in these Tables should be added.

5. A number of references in the bibliography are duplicated. For example, 24 and 32; 34 and 35; 71 and 77; 89 and 91.

6. Some important references on the detection of nitrobenzene and other nitroaromatic toxicants in environmental samples are strongly recommended to be added:https://doi.org/10.3390/chemosensors9090255; https://doi.org/10.1016/j.foodchem.2021.131279; https://doi.org/10.1016/j.cjac.2022.100215; etc.

Minor editing of English language required.

Author Response

Reviewer 3

  1. Figures in the text should be in the manuscript after they are mentioned in the text and not before the sections in which they are described.

We have verified and placed all the figures after they are mentioned in the texts.

  1. On page 5, section 2.5 breaks off in the middle of a sentence. References 50-52 were missed.

The missed text and references were added in the manuscript as per the reviewers’ comments. Please see line # 152.

  1. On page 7, reference 78 is mentioned before the corresponding references 75-77.

The reference was checked and sorted accordingly. Please check line # 245-260.

  1. There are no references to Tables 2-4 in the text. In addition, relevant comments regarding the data in these Tables should be added.

As per the reviewer suggestions, the reference and the comments to table 2-4 (now table 3-5) were added in the text. Please look line # 288, 422 and 479.

  1. A number of references in the bibliography are duplicated. For example, 24 and 32; 34 and 35; 71 and 77; 89 and 91.

The duplicated references in the manuscript were removed.

  1. Some important references on the detection of nitrobenzene and other nitroaromatic toxicants in environmental samples are strongly recommended to be added:https://doi.org/10.3390/chemosensors9090255; https://doi.org/10.1016/j.foodchem.2021.131279; https://doi.org/10.1016/j.cjac.2022.100215; etc.

As per the reviewer suggestions, the recommended references were added in the manuscript.

Round 2

Reviewer 1 Report

Although I am not an expert in the English language, I still see many errors in spelling and punctuation in the text.

Reviewer 3 Report

The authors have made an efficient revision on their work, which can be accepted in the current form.

Minor editing of English language required.